

# Pressure-dependent structural and electronic instabilities in LaSb$_2$

**Theodore I. Weinberger$^\star$, Christian K. de Podesta, Jiasheng Chen,
Stephen A. Hodgson and F. Malte Grosche**

Cavendish Laboratory, University of Cambridge, Cambridge CB3 0HE, United Kingdom

$\star$ tiw21@cam.ac.uk

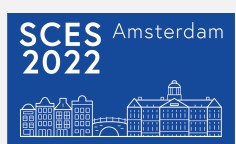

*International Conference on Strongly Correlated Electron Systems
(SCES 2022)
Amsterdam, 24-29 July 2022*

## Abstract

LaSb$_2$ exhibits a large, non-saturating, linear magnetoresistance at low temperatures, defying the expectations of Fermi liquid theory. This is thought to be caused by charge density wave order emerging below an abrupt, hysteretic anomaly in the resistivity at $\sim$ 355 K. We find that, under hydrostatic pressure, this anomaly becomes much more pronounced, develops strong hysteresis, and shifts rapidly towards lower temperatures. The anomaly is fully suppressed by only 6 kbar. Moreover, we observe a second transition anomaly at lower temperature, which likewise disappears under pressure. These findings are discussed in the context of a structural transition recently discovered in the sister material CeSb$_2$ and of density functional theory calculations, which indicate that the SmSb$_2$-type structure adopted by LaSb$_2$ at ambient conditions is unstable at moderate applied pressures.

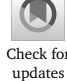
## 1 Introduction

The recent discovery of unconventional superconductivity in CeSb$_2$ beyond a pressure-induced structural transition [1], has prompted a re-examination of the rare-earth diantimonide family. The $R$Sb$_2$ materials crystallise in a series of highly-anisotropic, layered structures which host a variety of electronic and magnetic ground states [2–4]. The nature of this magnetic ordering is governed by the interplanar $R-R$ distance, making them ideal for tuning magnetism under pressure [5].

LaSb$_2$, the isostructural, non-magnetic counterpart to CeSb$_2$, exhibits superconductivity at ambient pressure [6]. While the magnetoresistance (MR) of a conventional Fermi-liquid metal is expected to scale quadratically with increasing magnetic field, the large MR of LaSb$_2$ remains linear, without saturating in fields in excess of 45 T [6, 7]. The observation of de

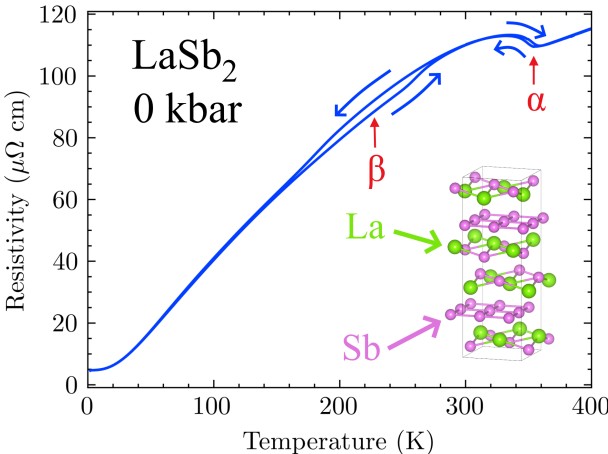

Figure 1: The temperature dependence of the resistivity of LaSb$_2$ is interrupted by two hysteretic anomalies; firstly, by an abrupt jump at $\sim 355\,$K, labelled $\alpha$; and secondly, by a wider hysteresis loop at around $250\,$K, labelled $\beta$. (inset) LaSb$_2$ adopts the SmSb$_2$ structure (space-group Cmca) under ambient conditions.

Haas van Alphen oscillations rules out explanations stemming from impurity effects [8]. It has been proposed that, instead, this anomalous MR is associated with a charge density wave (CDW) state, driven by strong Fermi surface nesting in LaSb$_2$, which develops at $\sim 355\,$K with a pronounced anomaly in the resistance, labelled $\alpha$ in Figure 1 [9]. The presence of CDW order in the closely related material LaAgSb$_2$ has been confirmed by X-ray measurements [10], but direct evidence of CDW order in LaSb$_2$ remains elusive so far [11,12].

Here, we explore the nature of the 355 K resistance anomaly in a series of high pressure measurements and by *ab initio* density functional theory (DFT) calculations.

## 2 Methods

Single crystals of LaSb$_2$ were grown by self flux from La (4N) and Sb (6N) in the ratio 1:9 and excess Sb was removed with a centrifuge [13]. The resulting crystals were formed in stacks of large plates ($\sim 10 \times 10$ mm), with the *c*-axis perpendicular to the plane. AC resistivity measurements were performed in the four-point configuration with I∥*ab*. Good electrical contact was ensured by spot-welding 25 $\mu$m Au wires to each sample and applying Dupont 6838 Ag paint. Pressure was generated using a BeCu/MP35N piston-cylinder cell (PCC) [14,15] with glycerol as the pressure-transmitting medium. The pressure inside the cell was determined from the superconducting transition temperature of a sample of Sn, while the temperature dependence of its measured resistivity was used to correct for thermal lag [16]. Measurements were recorded between 400 K and 1.76 K using the ACT module of the Quantum Design Physical Property Measurement System (PPMS).

Electronic structure calculations were performed using CASTEP [17]. Pseudopotentials were generated within the generalised gradient approximation [18] and calculations were converged on $15 \times 15 \times 5$ Monkhorst-Pack *k*-space grids [19]. These parameters reproduce the ambient pressure lattice parameters reasonably well (Table 1). The crystal structure of LaSb$_2$ was modelled under pressure by optimising the free energy as a function of both the lattice parameters and internal atomic positions. Within CASTEP this is done by using the Broyden–Fletcher–Goldfarb–Shanno algorithm [20], where the Hessian of atomic forces is calculated in the mixed space of internal and cell degrees of freedom. This allows simultaneous

Table 1: Ambient pressure lattice parameters of $LaSb_2$ obtained by X-ray diffraction (XRD) measurements are consistent with DFT results calculated using Perdew-Burke-Ernzerhof (PBE) pseudopotentials.

| Method | a/Å | b/Å | c/Å | Vol. per La/Å$^3$ |
|---|---|---|---|---|
| XRD (300 K) [3] | 6.32 | 6.17 | 18.57 | 90.55 |
| DFT | 6.39 | 6.27 | 18.62 | 93.31 |

optimisation of both lattice parameters and atomic coordinates. Further, CASTEP performs optimisation at fixed external stress allowing cell optimisation to be performed with varying external pressure.

## 3 Results

At ambient pressure, the resistivity of $LaSb_2$ exhibits two clear hysteretic anomalies: anomaly $\alpha$ at $\sim$ 350 K, and anomaly $\beta$ at $\sim$ 250 K (Figure 1). Anomaly $\alpha$ has only a small hysteresis on the order of 5 K, whereas anomaly $\beta$ exhibits significant hysteresis over a range on the order of 100 K. The hysteresis indicates that both of these transition are first order. These two anomalies can be distinguished by the order of the hysteresis in the resistivity.

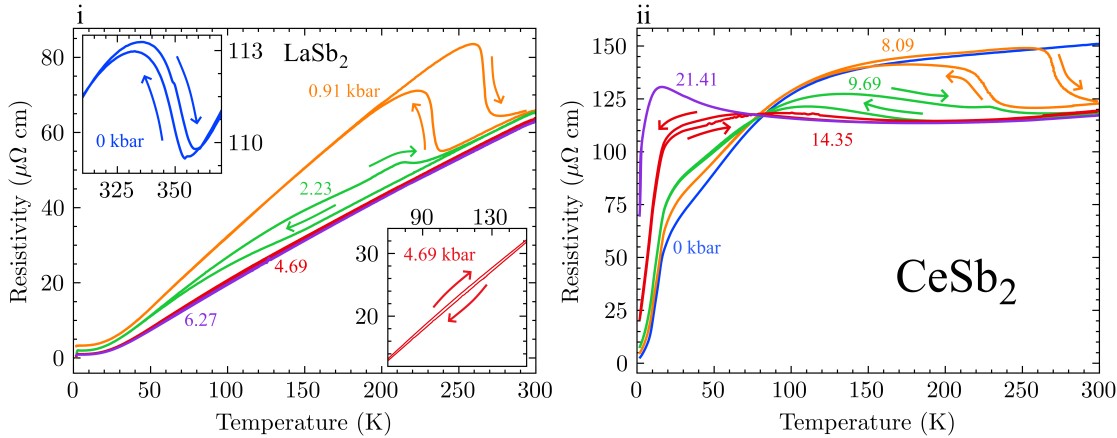

Figure 2: (i) The pressure dependence of the resistivity of $LaSb_2$ displays only a single, hysteretic anomaly above ambient pressure. This anomaly is rapidly suppressed, and disappears by $\sim$ 6 kbar. As this occurs, the magnitude of the change in resistivity decreases, whereas the width of the hysteresis loop increases. (inset left) Anomaly $\alpha$ exhibits only moderate hysteresis on the order of $\sim$ 5 K. (inset right) A small hysteresis loop is still visible at 4.69 kbar, after correcting for thermal lag. (ii) The resistivity of $CeSb_2$ under pressure also displays a hysteretic anomaly, which bears significant resemblance to the one seen in $LaSb_2$. This anomaly has been confirmed to correspond to a structural transition from the $SmSb_2$ ambient pressure structure to a different high pressure structure, most likely of the $YbSb_2$-type [1].

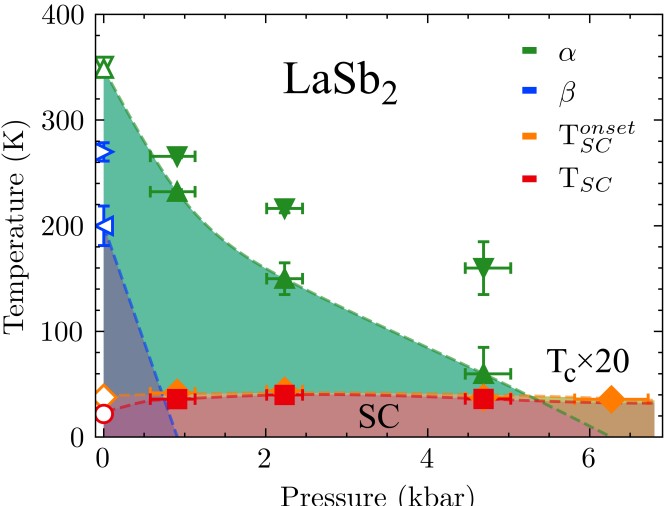

Figure 3: The phase diagram of $LaSb_2$ under pressure depicts the temperature dependence of the superconducting transition and the suppression of anomalies $\alpha$ and $\beta$. The superconductivity is initially enhanced, peaking at $\sim$4 kbar, before weakening with further increasing pressure. Filled markers correspond to data measured in a PCC, while empty markers represent data recorded at ambient pressure. A circular marker indicates $T_{SC}$ as measured by Galvis *et al.* [12]. The temperatures of anomalies $\alpha$ and $\beta$ on warming and cooling are marked with triangles pointing in different directions.

Even at the lowest applied pressure of $\simeq 1$ kbar, a single, pronounced resistivity anomaly replaces the two original anomalies, $\alpha$ and $\beta$, observed at ambient pressure (Figure 2 i). As the directions of the resistivity jumps when warming and cooling are the same as in the ambient pressure $\alpha$ anomaly, we identify this anomaly as a continuation of $\alpha$. However, the high pressure transition signature is highly hysteretic – like the $\beta$ transition at ambient pressure. The transition temperature is suppressed very rapidly from its ambient pressure value and disappears altogether at approximately 6 kbar. Alongside this suppression, increasing pressure reduces the magnitude of the resistance anomaly and increases the range of its temperature hysteresis. This increase in the hysteresis width as the transition is suppressed to lower temperatures may be attributed to kinetic effects associated with a strongly first order phase transition. However, additional contributions due to non-hydrostatic pressure conditions caused by the frozen pressure medium cannot be quantified at present. Similar results were recently presented in [21].

The superconducting transition temperature, $T_{SC}$, also depends on pressure (Figure 3). At ambient pressure, the transition is broad with a full superconducting state only reached below 1.2 K [12]. Increasing the pressure to $\sim$1 kbar brings the transition to above 1.7 K and significantly sharpens it. The pressure phase diagram indicates a maximum in $T_{SC}$ of $\sim$2.5 K at around 3-4 kbar. Further increasing the pressure then slowly suppresses $T_{SC}$.

## 4 Discussion

The shape and pressure dependence of the observed resistance anomaly are strongly reminiscent of the case of $CeSb_2$ (Figure 2 ii). In $CeSb_2$, high pressure x-ray diffraction (XRD) measurements confirm a pressure-induced structural transition from the $SmSb_2$-type struc-

ture to YbSb$_2$-type structure [1]. This prompts the question whether the observed resistivity anomalies in LaSb$_2$ might, likewise, be attributed to a structural instability rather than CDW order.

There has not yet been any direct measurement of CDW order in LaSb$_2$. The identification of the $\alpha$ anomaly as a signature of a CDW transition [21] primarily rests on the large linear MR observed in LaSb$_2$ at low temperature. However, CDW order is neither necessary nor sufficient to explain this phenomenon, as similar linear MR also arises in some Weyl and Dirac semi-metals [22–24] characterised by small Fermi surface pockets. Moreover, linear MR is observed at higher pressures even after anomaly $\alpha$ is fully suppressed [21]. CDW order has been reported in the (La/Ce)Sb$_2$ substitution series [9], but the study lacks direct observation of the CDW ordering wavevector by XRD, whereas the reported changes in bond lengths and lattice parameters might also be consistent with a structural transition. Furthermore, no evidence of a CDW ordering wavevector was found when directly probing LaSb$_2$ at low temperatures via scanning tunnelling microscopy [12].

It is notable that crystals of LaSb$_2$ grown by flux are predisposed towards twinning [6], which is often an indication that the crystal has been cooled through a structural transition, like in the case in the famous Dauphiné twins of quartz [25]. This is consistent with reports of a distinct, high-temperature form of LaSb$_2$ [26]. It has also been suggested that LaSb$_2$ is susceptible to a pressure-induced structural transition, in which the structure becomes more symmetric in the $ab$ plane, and the $c$-axis is compressed. This is indeed the case in CeSb$_2$ [6].

The crystal structures of the rare-earth diantimonides which can be grown at ambient pressure ($R =$La-Tb, Yb) consist of $R$-Sb bilayers, separated by a distorted square net of Sb [2,5]. The majority of the rare-earth diantimonides, including LaSb$_2$, crystallise in the orthorhombic SmSb$_2$ structure (space group Cmca, Figure 4 left). EuSb$_2$ and YbSb$_2$, however, both adopt higher symmetry variations of this structure, in which the neighbouring $R$-Sb layers are offset and pulled together by zig-zag chains of Sb atoms. The former adopts the P2$_1$/m space-group,

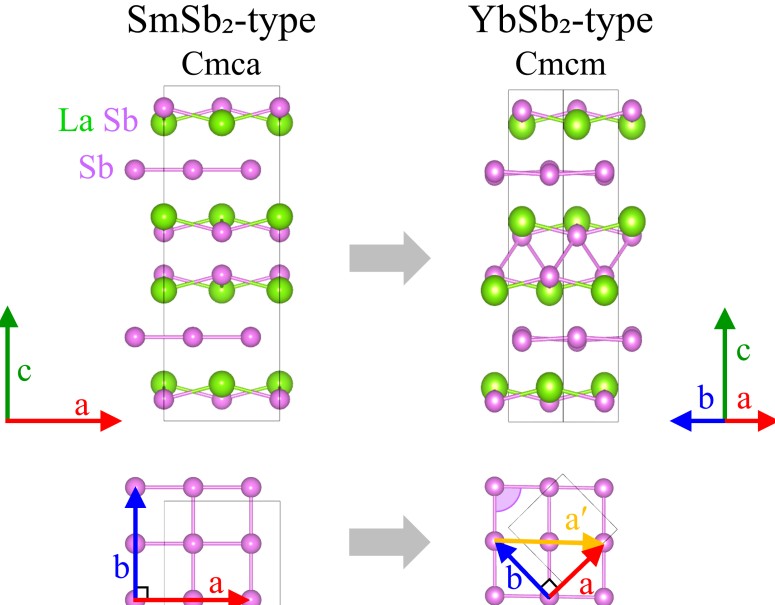

Figure 4: *Ab initio* calculations (see text) imply that LaSb$_2$, which adopts the SmSb$_2$ structure under ambient conditions (left), becomes unstable to the more symmetric YbSb$_2$ structure (right) under pressure. Structures generated using VESTA software package [27].

due to a distortion driven by the $Eu^{2+}$ valence, whereas in the latter, a structure with the Cmcm space-group is stabilised by the contribution of trivalent Yb on the predominantly $Yb^{2+}$ site [5]. As La does not support a divalent state, we expect it to undergo a transition to the $YbSb_2$-type structure under pressure, where the compressed layers and smaller volume reduce the free energy relative to the $SmSb_2$ structure [1].

In the transition from the $SmSb_2$-type structure to the $YbSb_2$-type structure, the planes of La and Sb atoms undergo a relative shear which allows the individual planes to come closer to each other. Consequently, the $c$-axis shortens, similar to the uncollapsed to collapsed tetragonal transition in some iron pnictides such as $CaFe_2As_2$ [28]. This behaviour is also seen in XRD reconstructed lattice parameters for the $SmSb_2$-to-$YbSb_2$ transition in $CeSb_2$ [1].

We have conducted extensive DFT calculations to investigate the possibility of a pressure-induced structural transition in $LaSb_2$ that matches the one observed in $CeSb_2$. These calculations suggest that at a surprisingly low pressure $\simeq 5$ kbar, the $SmSb_2$-type structure, adopted by $LaSb_2$ under ambient conditions, could become unstable to the $YbSb_2$-type structure. At ambient pressure, the $SmSb_2$-type structure is favoured as the ground state relative to the $YbSb_2$-type structure by $2.5 \times 10^{-3}$ eV/La. However, under pressure the larger volume of the $SmSb_2$-type structure relative to the $YbSb_2$-type structure means that it is rapidly destabilised. By 5 kbar the $YbSb_2$-type structure is energetically more favourable than the $SmSb_2$-type structure and remains the relative ground state structure up to at least 10 kbar. This is in good agreement with our high pressure experiments and supports the existence of a pressure-induced structural transition in $LaSb_2$.

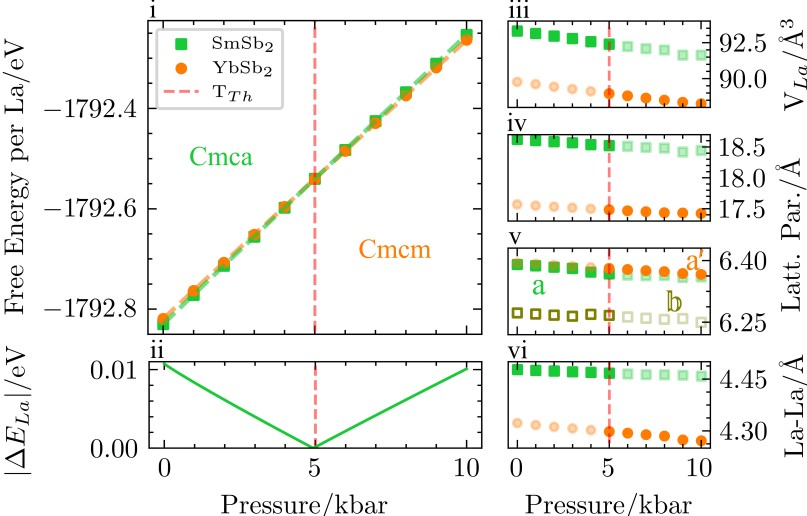

Figure 5: (i) The absolute energies of the $SmSb_2$-type structure and $YbSb_2$-type structure vary as function of pressure from 0 to 10 kbar. (ii) The absolute difference in energies between the $SmSb_2$-type structure and the $YbSb_2$-type structure indicates that there is a theoretical transition at 5 kbar. (iii) The volume of both cells per La atom demonstrates that the $YbSb_2$ has a smaller volume making it more favourable at high pressures. (iv) The $c$-axis of both structures shows how the transition from the $SmSb_2$-type structure to the $YbSb_2$-type structure corresponds to a collapse of the $c$-axis and shift in atomic planes. (v) The behaviour of the $a$ and $b$ lattice parameters in the $SmSb_2$-type structure and effective lattice parameter $a'$ in the $YbSb_2$-type structure indicate that the dominant compression is in the c-direction. (vi) The La-La separation in both structures is a further indicator of how the $SmSb_2$-to-$YbSb_2$ transition corresponds to a volume collapse.

# 5 Conclusion

The two high-temperature resistivity anomalies and the low temperature superconducting transition in ambient pressure $LaSb_2$ have been tracked under applied pressure. Anomaly $\alpha$ is rapidly suppressed to 0 K by $\sim 6$ kbar, whereas anomaly $\beta$ is not visible above ambient pressure. At present, these anomalies cannot be definitively assigned to either CDW ordering or a structural transition. Further work such as high pressure XRD will be required to determine whether the high pressure resistivity anomaly observed in $LaSb_2$ may be attributed to a structural transition similar to that observed in $CeSb_2$.

# Acknowledgements

We thank, in particular, O. Squire and G. Lampronti for helpful discussions and Z. Feng for early crystal growth work. The project was supported by the EPSRC of the UK (grants no. EP/K012894 and EP/P023290/1) and by Trinity College. This work was performed using resources provided by the Cambridge Service for Data Driven Discovery (CSD3) operated by the University of Cambridge Research Computing Service (www.csd3.cam.ac.uk), provided by Dell EMC and Intel using Tier-2 funding from the Engineering and Physical Sciences Research Council (capital grant EP/T022159/1), and DiRAC funding from the Science and Technology Facilities Council (www.dirac.ac.uk).

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
