# Peer review of "Pressure-dependent structural and electronic instabilities in LaSb2"

_SciPost Physics Proceedings, doi:SciPost Phys. Proc. 11, 018 (2023)_

## Round 1 · Referee Report · Anonymous (Referee 1) · 2022-11-29

Report

The manuscript by Weinberger et al. deals with LaSb2 under pressure. Transport measurements under pressure reveal the alpha anomaly is rapidly suppressed. A phase diagram, including a superconducting phase, is presented in Fig. 3. Electronic structure calculations predict a structural phase transition under pressure. A point of discussion in the paper is whether the observed phase transition is due to a CDW or purely structural.

This is a sound paper that deserved to be published in SciPost. I have two comments, but it is optional to take these into account in a revised version of the manuscript. 1. At the highest pressure, 6.27 kbar, the phase diagram suggests the alpha transition should still be present, but it is not. Could there be a structural transition between 4.69 and 6.27 kbar, such that the alpha transition (if a CDW) does not take place in the high-pressure structural phase? 2. I find it difficult to reconcile Fig. 5 (i) and (ii). Up to 5 kbar there is a clear difference between the red dots and green triangles, but between 5 and 10 kbar they seem to fall on top of each other.

---

## Editorial Decision

published